# Peer review of "Medication Regimen Complexity and Patient-Centred Outcomes in Patients Undergoing Peritoneal Dialysis"

_healthcare, 2024, doi:10.3390/healthcare12212121_

Round 1
Reviewer 1 Report
Comments and Suggestions for Authors
The manuscript (healthcare-3252670) investigated the effect of medication regimen complexity on patients under PD and evaluation of medication adherence and other patient outcomes. The authors may look into following comments which might help them to improve the manuscript:
· Authors should address the potential of selection bias as the recruitment of patients was done from single center.
· How would authors justify the selection of duration of recruitment which could allow more representative sample and avoid the potential for temporal biases in outcomes.
· Why did author not checked complexity and adherence associated with peritoneal dialysis treatments at home?
· Did authors adopt any specific approach in the study to deal with recall bias which is usually generated in the interviews based on patient self-reportting.
· The short-term antibiotics and non-oral medications were exluded from the medication list which could undermine the complexity of overall treatment regimen. How can authors justify such an exclusion.
· Why did authors select MGLS for evaluation in the study instead of more advanced scales, to account for variation in patient related outcomes due to adherence.
· Did authors checked the baseline HRQoL of participants prior to study? If yes, please include details.
· Authors should include discussion regarding the selection of the MRCI may affect the ability to compare between outcomes of different studies.
· Authors should provide detailed explanation about association between gender and phosphate levels with non-adherence.
· Discussion regarding crucial differences between PD and HD patients with regards to association between outcomes of HRQoL and adherence.
· The limitations of employing single time-point measurement of medication complexity should be thoroughly discussed as treatment regimen may require adjustments.
Author Response
We appreciate your time to review this manuscript. Please find the detailed responses below and the corresponding revisions/corrections highlighted in the re-submitted files.
Comments 1: Authors should address the potential of selection bias as the recruitment of patients was done from single centre.
Response 1: Although the study was conducted in a single dialysis centre, this centre is the largest peritoneal dialysis centre in Australia (Page 8, paragraph 2, line 273) with over 320 patients. The participants’ diverse background (Page 4, participants’ primary languages in Table 1) and the large sample size of n >100 compared to that of existing local literature (approximately n = 60) adds to the strength of the study. However, we have added this as one of the limitations and have made the following changes on page 8, paragraph 2, line 280 ‘Although this study was conducted in a large centre that caters to people from diverse backgrounds, this may not be representative of the entire Australian population’.
Comments 2: How would authors justify the selection of duration of recruitment which could allow more representative sample and avoid the potential for temporal biases in outcomes.
Response 2: All peritoneal dialysis patients at the centre were invited to participate in the study and the final number of participants were those that were interested and provided consent to participate. Hence, extending the duration of recruitment would not have affected the representative sample size.
Comments 3: Why did author not check complexity and adherence associated with peritoneal dialysis treatments at home?
Response 3: The authors agree that complexity and adherence associated with peritoneal dialysis treatment at home are important factors that affect patient outcomes. However, this was beyond the scope of our study as we only focused on assessing complexity on the medication aspect. Amendments made – Page 8, paragraph 2, line 297.
Comments 4: Did authors adopt any specific approach in the study to deal with recall bias which is usually generated in the interviews based on patient self-reporting.
Response 4: In addition to the self-report, we also collected the data from patients’ community pharmacies records of dispensing history (Page 2, paragraph 4, line 84). From the dispensing history, patients’ medication adherence could be indirectly ascertained and was further discussed during the patient interview.
Comments 5: The short-term antibiotics and non-oral medications were excluded from the medication list which could undermine the complexity of overall treatment regimen. How can authors justify such an exclusion.
Response 5: Non-oral medications were only excluded in the pill burden count but not in the main complexity measure of MRCI in the study (Page 2 paragraph 4, line 87 and page 3, line 102) as the authors would like to further investigate how the overall medication regimen and oral medications affected patient-related outcomes. Inclusion of short-term antibiotics for an acute infection, in the authors’ opinions, would lead to over-estimation of medication regimen complexity.
Comments 6: Why did authors select MGLS for evaluation in the study instead of more advanced scales, to account for variation in patient related outcomes due to adherence.
Response 6: The authors selected MGLS for evaluation in the study as it is a validated, reliable and widely used instrument in the clinical setting for difference chronic diseases including chronic kidney disease (Page 3, paragraph 2, line 112). Considering that the patient interview during the study included obtaining and/or confirming patients’ medication lists as well as assessing patients’ HRQoL using another set of questionnaires, the ease of implementation of MGLS in real practice lead to the use of this tool in our study.
Comments 7: Did authors check the baseline HRQoL of participants prior to study? If yes, please include details.
Response 7: The baseline HRQoL of participants of was not assessed prior to study as there were no interventions implemented to participants’ medication regimen which could potentially affect participants’ HRQoL.
Comments 8: Authors should include discussion regarding the selection of the MRCI may affect the ability to compare between outcomes of different studies.
Response 8: Although the lack of a standardised definition of medication regimen complexity would most likely account for the differences in findings between the studies, the use of different medication regimen complexity assessment methods and MRCI as the main measure in our study which includes comprehensive aspects of a medication regimen enabled us to compare between the outcomes of different studies. Amendments made – Page 8, paragraph 1, line 264
Comments 9: Authors should provide detailed explanation about association between gender and phosphate levels with non-adherence.
Response 9: The reasons of gender differences influencing medication adherence are multifactorial and could include sociodemographic, psychological and other factors, although similar findings from previous study suggested poorer outcomes in males with CKD due to lower adherence to treatment as one of the major causes.
The lower serum phosphate levels among adherent participants in our study was possibly due to better phosphate binder and/or dietary adherence. This was also shown in previous studies conducted among patients undergoing HD. Lower serum phosphate levels, in turn, found as a predictor of medication adherence in our study further indicate the importance of the relationship between both outcomes in patients treated with PD.
Amendments made – Page 7, paragraph 3, lines 237-241 & 247-251
Comments 10: Discussion regarding crucial differences between PD and HD patients with regards to association between outcomes of HRQoL and adherence.
Response 10: Although in general, patients receiving PD were shown to have better QoL and other clinical outcomes compared to patients receiving HD, the comparison of PD versus HD and the associated outcomes was not part of this study’s objectives. Amendments made – Page 7, paragraph 2, line 231
Comments 11: The limitations of employing single time-point measurement of medication complexity should be thoroughly discussed as treatment regimen may require adjustments.
Response 11: MRCI measured at a single timepoint may not fully capture the complexity, potentially leading to an over- or under-estimation of medication regimen complexity as patients with kidney failure frequently have their treatment adjusted to manage multiple comorbidities which can cause dynamic changes in their health status. Such limitation of a cross-sectional study design could be addressed in future studies by incorporating a longitudinal study design to gain better understanding of medication complexity among patients receiving PD. Amendments made – Page 8, paragraph 2, line 289
Reviewer 2 Report
Comments and Suggestions for Authors
Authors evaluated the Medication Regimen Complexity Index (MRCI) in patients undergoing peritoneal dialysis. Data on comorbidities, automated peritoneal dialysis (APD) modality, and adherence to treatment regimens were presented. Male gender, MRCI total score, serum phosphate level, EQ-5D-5L and EQ VAS were predictors of medication non-adherence in this study.
The authors noted that the self-reported adherence tool, although commonly used, may underestimate adherence and could impact our conclusions.
Additionally, the sample size was not large enough to assess the influence of underlying diseases like heart failure, hypertension, etc. on MRCI and medication adherence.
It is unclear if all excluded patients were on hemodialysis (HD) or receiving kidney supportive care/end of life care while on peritoneal dialysis (PD).
Was the history of using herbal supplements also included in your study. If so please specify.
For better understanding, provide an example of how MRCI was calculated.
Please discuss the strengths and limitations of your study.
Author Response
We appreciate your time in reviewing this manuscript. Please find the detailed responses below and the corresponding revisions/corrections highlighted in the re-submitted files.
Comments 1: The authors noted that the self-reported adherence tool, although commonly used, may underestimate adherence and could impact our conclusions.
Response 1: The authors selected MGLS for evaluation in the study as it is a validated, reliable and widely used instrument in the clinical setting for difference chronic diseases including chronic kidney disease (Page 3, paragraph 2, line 113). In addition to the self-report, we also collected the data from patients’ community pharmacies records of dispensing history (Page 2, paragraph 4, line 84). From the dispensing history, patients’ medication adherence could indirectly be ascertained through the frequency of medication refill and this was further discussed during the patient interview if required. Moreover, considering that the patient interview during the study included obtaining and/or confirming patients’ medication lists as well as assessing patients’ HRQoL using another set of questionnaires, the ease of implementation of MGLS in real practice led to the use of this tool in our study. Objective assessment of random serum phosphate level closest to the interview date was also obtained to further confirm medication adherence (Page 3, paragraph 2, line 116).
Comments 2: Additionally, the sample size was not large enough to assess the influence of underlying diseases like heart failure, hypertension, etc. on MRCI and medication adherence.
Response 2: Considering the comparison of the number of patients undergoing dialysis versus patients with other chronic illnesses such as heart failure and hypertension, our study had a larger sample size (n >100) compared to that of existing local literature (approximately n = 60). This was also the first study in Australia conducted among PD patients at the largest dialysis centre in Australia (Page 8, paragraph 2, line 272) and participants of diverse backgrounds were included in the study (Page 4, participants’ primary languages in Table 1).
Comments 3: It is unclear if all excluded patients were on haemodialysis (HD) or receiving kidney supportive care/end of life care while on peritoneal dialysis (PD).
Response 3: We excluded patients on temporary HD and those receiving kidney supportive care. The exclusion criteria are enumerated on Page 2, paragraph 2, line 70.
Comments 4: Was the history of using herbal supplements also included in your study. If so please specify.
Response 4: Herbal supplements were not included in the study. Amendments made in Page 2, paragraph 3, line 87.
Comments 5: For better understanding, provide an example of how MRCI was calculated.
Response 5: An explanation of MRCI calculation is given in Page 2, paragraph 5, line 95.
Comments 6: Please discuss the strengths and limitations of your study.
Response 6: Strengths and limitations have been discussed on Page 8, paragraph 2, lines 277-294.
Reviewer 3 Report
Comments and Suggestions for Authors
Journal: Healthcare (ISSN 2227-9032)
Manuscript ID: healthcare-3252670
Type: Article
Title: Medication regimen complexity and patient-centred outcomes in patients undergoing peritoneal dialysis
Authors: Jing Xin Goh , Kamal Sud , Wubshet Tesfaye , Connie Van , Shrey Seth , Surjit Tarafdar , Ronald L Castelino
Section: Medication Management
Special Issue: Medication Therapy Management in Healthcare
The above-mentioned manuscript is exploring and quantifying medication regimen complexity and evaluating patient-centered outcomes including medication adherence and its determinants in patients undergoing peritoneal dialysis. Results indicate that over 90% of patients experiences polypharmacy. Participants who were adherent to their prescribed medications had a higher EQ-5D-5L and the EQ VAS score, serum phosphate level among adherent participants was shown to be lower. Regarding the predictors of medication non-adherence authors found that male participants were more likely to be non-adherent as well as participants with higher serum phosphate level.
The manuscript fits with the aims of Healthcare journal and a special issue Medication Therapy Management in Healthcare, however, there are some minor shortcomings and some modifications should be carried out to improve the manuscript.
Specific comments:
Introduction: Informative, appropriate length, and well written. Although only 5 references were cited, 3 of them are more recent than 5 years.
Material and methods:
To the best of my knowledge, section materials and methods is well written, systematical, following Instructions to Authors. Study design is well explained. Authors have applied clear and correct inclusion and exclusion criteria for patients involved. The authors have used widely accepted and adequate tests (scales) to assess the outcomes. Applied statistical methods were adequate for this kind of data.
Comment 1: In section 2.1. Study design and 2.2. Measures (variables) there are initials of the research pharmacist and renal pharmacist, which I find non-necessary. It may be mentioned in the Authors contributions section of the manuscript. To be clear – terms “renal pharmacist” and “ research pharmacist” may stay in the text, just remove the initials, while in Authors contribution section, add terms “renal” and “research” pharmacist along with names of authors.
Comment 2: explore the possibility to use more up to date references, since only 3 of 9 used references are published over previous 5 years.
Results
Overall, the results are well presented, statistical differences are adequately stated in the text and in table legends.
Comment 1: One could think that current alcohol consumption would affect the outcomes and medication non-adherence. So my question is, was alcohol consumption separately analyzed for former and current or was that analyzed as altogether factor.
Discussion
The discussion is purposeful, clearly conducted and the authors have critically reviewed the data obtained.
Comment 1: I advise authors to be careful with the statement of being “the first study assessing medication regimen complexity, medication adherence as well as patient-centered outcomes focusing on patients receiving PD.” I have not conveyed that detailed search, but if they stick to this statement, it must be triple checked.
Comment 2: Authors have cited 9 references in the section Discussion, and just 3 of them are published in last 5 years. The reference list must be enriched with more articles and preferably not older than 5 years. Include both the articles that support your results, but also the articles which report different results, with an explanation regarding different results compared to your study.
REFERENCES
A percentage of references older than 5 years is a bit higher (16, compared to 9 published in last 5-6 years) and this must be corrected particularly in Introduction and Discussion sections.
Author Response
We appreciate your time to review this manuscript. Please find the detailed responses below and the corresponding revisions/corrections highlighted in the re-submitted files.
Material and methods:
Comments 1: In section 2.1. Study design and 2.2. Measures (variables) there are initials of the research pharmacist and renal pharmacist, which I find non-necessary. It may be mentioned in the Authors contributions section of the manuscript. To be clear – terms “renal pharmacist” and “ research pharmacist” may stay in the text, just remove the initials, while in Authors contribution section, add terms “renal” and “research” pharmacist along with names of authors.
Response 1: Thank you for pointing that out, amendments have been made – Page 2, paragraph 3, line 89.
Comments 2: Explore the possibility to use more up to date references, since only 3 of 9 used references are published over previous 5 years.
Response 2: Newer references involving the use of MGLS to evaluate medication adherence in chronic kidney disease have been added. However, the other references that were published over 5 years included the original articles for:
- MRCI – a validated tool increasingly being used to assess medication regimen complexity
- Electronic data capture and coding tool of MRCI developed to compute the total MRCI scores
- 4-item MGLS – a long-established, validated and widely used instrument in clinical settings to evaluate medication adherence
- EQ-5D-5L and EQ VAS – one of the most widely used tools to measure the HRQoL of patients in clinical trials and real-world clinical settings.
Results:
Comments 3: One could think that current alcohol consumption would affect the outcomes and medication non-adherence. So my question is, was alcohol consumption separately analysed for former and current or was that analysed as altogether factor.
Response 3: Former and current alcohol consumption was analysed as an altogether factor. Amendments have been made in Page 4, Table 1 for clarification.
Discussion:
Comments 4: I advise authors to be careful with the statement of being “the first study assessing medication regimen complexity, medication adherence as well as patient-centred outcomes focusing on patients receiving PD.” I have not conveyed that detailed search, but if they stick to this statement, it must be triple checked.
Response 4: This has been repeatedly checked. There were only two studies from Australia assessing medication regimen complexity and patient outcomes in patients with CKD. One was conducted in adults with pre-dialysis CKD and the other among patients receiving HD. To date, no study in Australia has been conducted to assess medication regimen complexity and its outcomes in patients undergoing PD.
Comments 5: Authors have cited 9 references in the section Discussion, and just 3 of them are published in last 5 years. The reference list must be enriched with more articles and preferably not older than 5 years. Include both the articles that support your results, but also the articles which report different results, with an explanation regarding different results compared to your study.
Response 5: Edits have been made to include more recent articles in the Discussion section. Nevertheless, the volume of research conducted in kidney diseases is significantly less compared to other chronic diseases such as diabetes and cardiovascular diseases. Studies assessing MRCI in patients with chronic kidney disease and kidney failure are limited, more so specifically to patients receiving peritoneal dialysis and hence the reason why many of the articles cited were older than five years and the need of this study.
References:
Comments 6: A percentage of references older than 5 years is a bit higher (16, compared to 9 published in last 5-6 years) and this must be corrected particularly in Introduction and Discussion sections.
Response 7: Addressed as per previous comments.
Reviewer 4 Report
Comments and Suggestions for Authors
The presented study, a combination of a retrospective analysis of baseline data and a prospective evaluation of patient-related outcomes, aimed to quantify medication regimen complexity and evaluate patient-centred outcomes including medication adherence and its determinants in patients undergoing PD. Medication regimen complexity was assessed using the Medication Regimen Complexity Index (MRCI), while patient outcomes were evaluated with validated self-reported questionnaires. It revealed that over half of the participants were adherent to their prescribed medications and that male participants were more likely to be non-adherent to medications compared to 30 female ones. In addition, patients on PD who were adherent to their medications had significantly better outcomes in terms of HRQoL and serum phosphate levels compared to non-adherent patients.
From a clinical practice's point of view, this is an interesting and useful study, emphasizing some important aspects of these complex patients although, on the other hand, there are not many new findings presented. It is, in general, a well designed, transparent and written study.
However, there are some issues that must be addressed and/or better explained, such as:
1. Introduction
1st paragraph
................ In Australia, the utilisation of PD has remained relatively stable over the last few years despite the COVID-19 pandemic in 2020 .............
- Comment: authors should explain in more detail what a possible influence of COVID-19 pandemic on utilisation of PD could be.
2. Results
Table 1 and 2:
Table 1 and 2 present results that are already presented in text. This is doubling of data presentation and should be avoided. Data should be presented either in text or in tables but not both.
3. Characteristics of study participants
Outcomes and determinants of medication non-adherence
paragraph 1:
........... Interestingly, the objective assessment of serum phosphate level among adherent participants was shown to be lower compared to non-adherent participants ..................
- Comment: authors present above statement as an interesting finding. However, it is expected that adherent participants have lower serum phosphate level compared to non-adherent participants because of more regular taking of phosphate binders as well as due to better adherence to low-phosphate diet.
4. Discussion
1st paragraph
.......... The adherence rate among participants in this study (65.5%) was also shown to be slightly better compared to that reported among facility-based HD patients (≤50%) [15,16], albeit previous literature indicated the non-adherence rate in PD patients varied from 13% to 50% [17]. Differences in adherence assessment tools, patient demographics, healthcare systems and settings may contribute to such discrepancies.
- Comment: it is expected that PD patients have higher adherence rate than facility-based HD patients because PD patients are generally younger, in better general medical condition and prepared to give more input in their treatment: If this has not been the case, they would not be able to deal with challenges of independent home PD treatment that demand more input than in case of facility-based HD. Author should discuss more about this.
Author Response
We appreciate your time to review this manuscript. Please find the detailed responses below and the corresponding revisions/corrections highlighted in the re-submitted files.
Introduction:
Comments 1: “................ In Australia, the utilisation of PD has remained relatively stable over the last few years despite the COVID-19 pandemic in 2020 .............”
Authors should explain in more detail what a possible influence of COVID-19 pandemic on utilisation of PD could be.
Response 1: Amendments made in Page 1, paragraph 1 to remove this statement from the manuscript.
Results:
Comments 2: Table 1 and 2 present results that are already presented in text. This is doubling of data presentation and should be avoided. Data should be presented either in text or in tables but not both.
Response 2: The authors believe that the results presented in text on top of the data in the tables allow readers to better understand the key findings and significant trends emphasised by the authors.
Comments 3: “........... Interestingly, the objective assessment of serum phosphate level among adherent participants was shown to be lower compared to non-adherent participants ..................”
Authors present above statement as an interesting finding. However, it is expected that adherent participants have lower serum phosphate level compared to non-adherent participants because of more regular taking of phosphate binders as well as due to better adherence to low-phosphate diet.
Response 3: Thank you for pointing that out, amendments made in Page 6, paragraph 1, line 185 to explain the findings as 'unsurprisingly'.
Discussion:
Comments 4: “.......... The adherence rate among participants in this study (65.5%) was also shown to be slightly better compared to that reported among facility-based HD patients (≤50%) [15,16], albeit previous literature indicated the non-adherence rate in PD patients varied from 13% to 50% [17]. Differences in adherence assessment tools, patient demographics, healthcare systems and settings may contribute to such discrepancies.”
It is expected that PD patients have higher adherence rate than facility-based HD patients because PD patients are generally younger, in better general medical condition and prepared to give more input in their treatment: If this has not been the case, they would not be able to deal with challenges of independent home PD treatment that demand more input than in case of facility-based HD. Author should discuss more about this.
Response 4: Amendments made in Page 7, paragraph 2, line 219 to include the elaborations as suggested by reviewer that the better adherence among PD patients could be attributed to several reasons as patients receiving PD are generally younger, more educated, and have fewer comorbidities compared to patients treated with HD. As a result, PD patients would be more independent in dealing with the challenges of managing their own treatment at home and hence the better motivation and commitment to their care plan.
Round 2
Reviewer 1 Report
Comments and Suggestions for Authors
The authors have thoroughly addressed the comments, and the modifications made to several sections have improved the manuscript. It is now suitable for final publication.